# Habitat Specificity, Host Plants and Areas of Endemism for the Genera-Group *Blepharida* s.l. in the Afrotropical Region (Coleoptera, Chrysomelidae, Galerucinae, Alticini)

**DOI:** 10.3390/insects12040299

**Published:** 2021-03-29

**Authors:** Mattia Iannella, Paola D’Alessandro, Walter De Simone, Maurizio Biondi

**Affiliations:** Department of Health, Life, and Environmental Sciences, University of L’Aquila, 67100 Coppito-L’Aquila, Italy; mattia.iannella@univaq.it (M.I.); walter.desimone@graduate.univaq.it (W.D.S.); maurizio.biondi@univaq.it (M.B.)

**Keywords:** *Afroblepharida*, *Blepharidina*, *Calotheca*, Anacardiaceae, Burseraceae, host plants, ecoregions, areas of endemism, sub-Saharan Africa

## Abstract

**Simple Summary:**

Knowledge of the processes that generate biodiversity is a core-issue of any conservation strategy because it allows predicting the effects of environmental changes in the number and distribution of target taxa. Some phytophagous insects can be good potential indicators of such processes, thanks to their wide distribution and their sensitivity to climate change, due to the association with specific environments and host plants. Unfortunately, this ecological information is often lacking. However, statistical tools allow reconstructing the ecological features of interest, based on the presence–absence data of the taxa, the climatic and vegetational features of their distributional areas, and the available data about their host plants. In this paper, we apply some geostatistical methods to identify processes and patterns of biodiversity at a continental scale, focusing on a group of phytophagous insects widespread in sub-Saharan Africa.

**Abstract:**

The genus *Calotheca* Heyden (Chrysomelidae) is mainly distributed in the eastern and southern parts of sub-Saharan Africa, with some extensions northward, while *Blepharidina* Bechyné occurs in the intertropical zone of Africa, with two subgenera, *Blepharidina* s. str. and *Blepharidina*
*(Afroblepharida)* Biondi and D’Alessandro. These genera show different ecological preferences. Through an up-to-date presence–absence dataset, in the light of the terrestrial ecoregions of sub-Saharan Africa and the distribution of their possible host plants, we interpreted the pattern of occurrence of these three supraspecific taxa, by geostatistical analyses in GIS and R environments. The separation of *Blepharidina* from *Calotheca* was probably driven by changes in climate as adaptation to more xeric and warm environments with a major occupancy of semidesert and savannah habitats, especially in the *Afroblepharida* species. Based on our data and analyses, *Calotheca* is mainly associated with *Searsia* (Anacardiaceae), and *Blepharidina* is likely associated with *Commiphora* (Burseraceae). This hypothesis is also corroborated by the widespread and even dominance of the *Commiphora* plants in the ecoregions where both *Blepharidina* s.str. and, above all, *Afroblepharida*, are more common. The main areas of endemism of the two genera are also differently located: *Calotheca* in the temperate zone; *Blepharidina* within the intertropical belt.

## 1. Introduction

The Alticini are a tribe of Coleoptera Chrysomelidae comprising over 534 genera and about 8000 species [1,2], occurring all over the world. Members of this tribe are commonly defined as “flea beetles” because of the presence of a metafemoral extensor tendon that enables them to jump [3,4,5]. Adult and larval stages mainly feed on stems, leaves, or roots, although rarely on flowers, of almost all the higher plant families, generally with high levels of specialization and in different environments [6,7,8]. The highest species richness occurs in the tropics of the southern hemisphere, even though our knowledge about the Alticini is still incomplete for these areas [2,9,10,11].

For a long time, *Blepharida* Chevrolat, 1836 [12] has been considered a widespread flea beetle genus occurring in the Nearctic, Neotropical, and Afrotropical regions and, to a small extent, in the Palearctic region, with the species *C. sacra* reported in Israel [10,13,14]. Recently, the species from sub-Saharan Africa, Asia Minor, and the Arabian Peninsula were attributed to the genera *Calotheca* Heyden, 1887 [15] and *Blepharidina* Bechyné, 1968 [16,17,18], confining the distribution of the true *Blepharida* to the Nearctic and Neotropical regions.

*Calotheca* shows a geographic distribution that includes the greater part of sub-Saharan Africa, with extensions in Israel and Saudi Arabia (Figure 1). Although it is absent in the north-western areas, it is particularly common in the eastern and southern parts of its distribution range [17]. Thirty-two species are currently attributed to this genus [17,19,20].

*Blepharidina* comprises two subgenera, *Blepharidina* s. str., hereafter *Blepharidina*, and *Blepharidina (Afroblepharida)* Biondi and D’Alessandro, 2017, hereafter *Afroblepharida* [17]. This genus occurs in the intertropical area of Africa with at least seventeen species of *Blepharidina*, mostly distributed in the south, and twelve species of *Afroblepharida* occurring largely in the central eastern area, including the Socotra island, with some species towards the north and one in the west [18,21,22]. The ranges of these two subgenera show an overlap in southern Kenya [8,17] (Figure 1).

Regarding the life cycle, general information is mainly available for *Blepharida* s.l. [23,24]. Generally, adult females lay eggs on their host stems during the summer; larvae are ectophytic and feed for approximately 20 days, then they eventually drop off the plant and find new shelter in the soil, where they pass through their pre-pupal and pupal stages. At the end of this phase, the adult emerges around the end of spring [25]. Molecular studies have confirmed that *Blepharida* s.l. species and their host plants, mainly represented by Anacardiaceae and Burseraceae, evolved their traits in response to mutual selective pressures [26].

The ecology of the Afrotropical *Afroblepharida*, *Blepharidina*, and *Calotheca* is very poorly known, including few with definitive host associations. The available records of *Calotheca* host plants identify the genus *Searsia* (generally reported as *Rhus*) from the Anacardiaceae family as primary host plants [23,27]. In this regard, it is important to clarify that the genus *Rhus* L. has traditionally been defined to include up 250 species worldwide. However, most of the African members of *Rhus* are now widely recognized to belong in the segregate genus *Searsia* F.A. Barkley, which is distributed mainly in southern Africa, as well as in Sicily and the Middle-East to Yunnan, with over 100 species [28,29,30]. Other genera of Anarcadiaceae, such as *Ozoroa* Delile and *Schinus* L., and the genus *Commiphora* Jacq. from the Burseraceae, are generically recorded as host plants for Blepharidini in Africa [31,32].

We tried to understand if and how climatic conditions have influenced the current distributions of *Afroblepharida*, *Blepharidina* and *Calotheca* in the Afrotropical region. We used ecological niche modelling, a technique applied for climate-related issues to many other case studies on Chrysomelidae [33,34,35,36], to highlight possible differences in their habitat preferences [8]. Our analysis focused on the climatic conditions, particularly those related to temperature and precipitation patterns, because of the significant influence of these parameters on the different types of vegetation and, consequently, on the distribution of these phytophagous beetles. Our models suggested modifications during the time of the suitable areas of *Afroblepharida*, *Blepharidina* and *Calotheca* led by changes in climate; specifically, the increase in xeric and warm environmental conditions.

Starting from this information, in this research we aim at: (i) interpreting the pattern of occurrence for all known species of *Afroblepharida*, *Blepharidina* and *Calotheca*, through a reliable and up-to-date presence–absence dataset, in the light of the terrestrial ecoregions reported for sub-Saharan Africa [37] and distribution of their possible host plants; (ii) correlating the distribution of *Calotheca* with the distribution of the *Searsia* plant genus; (iii) hypothesizing the possible host plants for *Afroblepharida* and *Blepharidina* using geostatistical analyses; and (iv) identifying and delimiting eventual areas of endemism.

## 2. Materials and Methods

### 2.1. Study Area and Flea Beetle and Plant Datasets

The study area includes sub-Saharan Africa, Madagascar, and the Arabian Peninsula and its adjacent northern areas representing the northern limit of the distribution of the taxa considered here (Figure 1). The datasets for the flea beetles *Afroblepharida* (69 occurrence localities), *Blepharidina* (162 occurrence localities), and *Calotheca* (797 occurrence localities), for a total of 61 species, were generated using both known localities and original data [17,18,19,20,21,22]. Data refer mainly to material preserved in the world’s main museums [17] and private collections and cover over a century. Part of the *Calotheca* from the Republic of South Africa was collected during expeditions carried out from 1990 to 2010, making it possible to obtain secure information on their host plants. *Afroblepharida* and *Blepharidina*, in contrast, have not been the target of recent collecting expeditions, also considering the difficult accessibility because of the political instability of many of the regions included in their distribution area. This is why for the species of *Afroblepharida* and *Blepharidina*, the available distributional and ecological data derive exclusively from the study of preserved material [18,21]. The datasets of the plants *Searsia* (13,605 occurrence localities) with 104 species and *Commiphora* (6343 occurrence localities) with 142 species, occurring in the African continent and Madagascar, were obtained from the GBIF portal (*Commiphora* accessed on 27 January 2021 [38]; *Searsia* accessed on 13 January 2021 [39]) and the available literature and checklists of African countries. The spatial information from the IUCN Red List (https://www.iucnredlist.org) (accessed on 27 January 2021) was also used to compare the occurrence localities such obtained with species and genus range data. All duplicate, doubtful, or low-precision records, as well as data missing some information in their attribute tables, were discarded from the analyses.

### 2.2. Ecoregions

Terrestrial ecoregions of the world (869, last update) were proposed by Olson et al. (2001) [37]. They are relatively large land units containing distinct assemblages of natural communities and species, with boundaries that approximate the original extent of natural communities prior to major land-use change. Ecoregions are classified into 14 major habitats, such as forests, grasslands, or deserts, and represent a useful framework for conducting biogeographical or macroecological research, for identifying areas of outstanding biodiversity and conservation priority, for assessing the representation and gaps in conservation efforts worldwide, and for communicating the global distribution of natural communities on earth. The shapefile used by us, including 132 ecoregions for Africa, Madagascar, and Arabian Peninsula (Appendix A), was downloaded from http://www.fao.org/land-water/land/land-governance/land-resources-planning-toolbox/category/details/en/c/1036295/ (accessed on 27 January 2021).

### 2.3. GIS Analyses

#### 2.3.1. Density of Species Richness and Occurrence

Occurrence localities for each target taxa were processed through the Point Density tool in ArcMap 10.0 [40], to obtain the magnitude of the occurrence density surface based on the neighborhood of localities. Additionally, the Calculate Richness and Endemicity process (tessellation resolution = 100 km) from the SDMtoolbox 2.4 ArcMap toolbox [41] was applied to the occurrence localities of both target species and plants, taking the species as the selected variable to perform the analysis. To obtain continuous responses of Species Richness over the study area, the outputs of the tool were first converted to point features, and then processed through the Kernel Density tool (output cell size = 1 km) in ArcMap 10.0 [40].

#### 2.3.2. Habitat Specificity

To determine the magnitude of a patch contributing to the overall richness at a vast scale, we applied the Habitat Specificity index [42] to the *Afroblepharida*, *Blepharidina* and *Calotheca* occurrences as a source of species’ information, using the ecoregions reported for sub-Saharan Africa [37] as target landscape patches. Considering that the latter are shaped following ecological boundaries (thus, in a non-standardized pattern), we excluded the “area” contribution of the formula by adopting the Halvorsen and Edvardsen correction [43], as suggested [42,43] and applied [44] in other cases. The corrected Habitat Specificity (S) of a patch (in this case, a single ecoregion) was thus calculated as Sij = ∑ (mj /ni), with m representing the number of species occurring in the jth patch, and n being the number of patches in which the ith species occurs, referring to the total landscape.

#### 2.3.3. Flea Beetle–Plant Species Associations

To assess the possible associations of *Afroblepharida* and *Blepharidina* with *Commiphora* plants, as well as to confirm those existing between *Calotheca* with *Searsia* plants, we calculated the spatial correlation between their respective kernel density rasters for Species Richness (obtained through the SDMtoolbox software, see above). The Band Collection Statistics tool in ArcMap 10.0 [40] was used for this aim. Moreover, 19 climatic variables were downloaded from the Worldclim.org repository ver. 2.1, at 30 arc-seconds spatial resolution [45]. Their averages (Table 1), calculated for all occurrences for every taxa, were used to carry out a cluster analysis (Euclidean distance and WPGMA clustering method) among *Afroblepharida*, *Blepharidina*, and *Calotheca* flea beetles, and *Searsia* and *Commiphora* plants, to evaluate the possible associations between the insects and the relative host plants. Statistical analyses were performed using NCSS11 software (NCSS, LLC, Kaysville, UT, USA). Finally, to detect the possibility of a niche overlap driven by climate, we evaluated the target flea beetle and plant occurrences by using the “hyperoverlap” package [46] in R environment [47]. This tool permits the evaluation of overlap or divergence between point datasets, such as in our case, using support vector machines to find the best classification (linear or polynomial), giving a classification matrix, for any n-dimensional set of point attributes [46]. The “hyperovelap_detect” function, which has proven to be resistant to sampling biases and to small numbers of points [46], was used to find climatic niche overlap for the aforementioned pairs, while the “hyperovelap_lda” function was used to obtain the three-dimensional plots (reporting a combined linear discriminant analysis, PCA 1 and PCA 2 residuals) resulting from the classification. This last function was needed because four of the Worldclim climatic variables were used for this analysis, namely, BIO1 (mean annual temperature), BIO7 (temperature annual range), BIO14 (precipitation of the driest month) and BIO18 (precipitation of the warmest quarter), identified to be the most relevant variables from the species distribution models built by D’Alessandro et al. [8], shared among *Afroblepharida*, *Blepharidina* and *Calotheca*.

#### 2.3.4. Areas of Endemism

To identify the areas of endemism for *Afroblepharida*, *Blepharidina*, and *Calotheca*, we used the Geographical Interpolation of Endemism (GIE) toolbox [48] for ArcMap 10.0. This method is independent of grid cells and is based on estimating the overlap between the distribution of species through a kernel spatial interpolation of centroids of species distribution and areas of influence. The latter is defined as the distance between the centroid and the farthest point of occurrence of each species. In this study, two categories of endemic species were considered: species with up to 100 km (class 1) and 300 km (class 2) of distance between the centroid and the farthest point. Only areas with at least two synendemic species were generally considered.

## 3. Results

### 3.1. Habitat Specificity and Flea Beetle-Plant Species Associations

The application of the Habitat specificity index identified the most important ecoregions that characterize the distribution of *Afroblepharida* (Figure 2), *Blepharidina* (Figure 3) and *Calotheca* (Figure 4) in terms of the magnitude that each ecoregion had in contributing to the overall richness. *Afroblepharida*, with its 12 species, clearly shows more xeric preferences. It occupies 13 different ecoregions, with over 75% of the area occupied by the “Northern *Acacia*-*Commiphora* bushlands and thickets” (31.2%), “Somali *Acacia*-*Commiphora* bushlands and thickets” (31.2%), and “Sahelian *Acacia* savanna” (14.4%) ecoregions. For the 17 species of *Blepharidina*, 20 ecoregions were identified, but over 60% of the total area is represented by the “Northern *Acacia*-*Commiphora* bushlands and thickets” (30.9%), “Angolan Miombo woodlands” (15.2%), and “Western Congolian forest-savanna mosaic “(15.2%) ecoregions. The coverage in ecoregions resulting for the 32 species of *Calotheca* was distinctly different. It includes 46 ecoregions, and about 42% of its extension is represented by only two ecoregions named “Drakensberg montane grasslands, woodlands and forests” (24.0%) and “Southern Africa bushveld” (17.9%).

The results of raster correlation between the point densities (Figure 5 and Figure 6), obtained for the occurrence localities for each taxon, gave a high correlation (Pearson’s *r*) for the *Calotheca*–*Searsia* pair (*r* = 0.676), as well as for the *Afroblepharida*–*Commiphora* (*r* = 0.576) and *Blepharidina*–*Commiphora* (*r* = 0.594) pairs. On the other side, the opposite pairs (*Afroblepharida*–*Searsia*, *Blepharidina*–*Searsia* and *Calotheca*–*Commiphora*) showed significantly lower *r* scores (0.07, 0.03, and 0.55, respectively).

Moreover, the cluster analysis (Figure 7) also gave clear results about the association of *Calotheca* with the *Searsia* plants and *Afroblepharida*/*Blepharidina* with the *Commiphora* plants. The kernel densities obtained from the Species Richness inferred through SDMtoolbox also resulted in a high correlation for *Afroblepharida*–*Commiphora* (*r* = 0.776), *Blepharidina*–*Commiphora* (*r* = 0.778) and *Calotheca*–*Searsia* (*r* = 0.929) pairs. The other correlations resulted in lower Pearson’s coefficients (*Afroblepharida*–*Searsia*, *r* = 0.221; *Blepharidina*–*Searsia*, *r* = 0.112; *Calotheca*–*Commiphora*, *r* = 0.557).

Beyond the magnitude that each ecoregion had in contributing to the overall richness, the climatic niche resulted as overlapping for *Afroblepharida*–*Commiphora*, *Blepharidina*–*Commiphora* and *Calotheca*–*Searsia* pairs (Figure 8); the misclassified points (i.e., the points which could not be assigned to one taxon only of each pair, because of the niche overlap) for the target flea beetles were 66.7% for *Afroblepharida*–*Commiphora* (number of support vector machines = 71), 38.7% for *Blepharidina*–*Commiphora* (number of support vector machines = 208), and 99.9% for *Calotheca*–*Searsia* (number of support vector machines = 1369) pairs. Instead, no overlaps were found for the *Afroblepharida*–*Searsia*, *Blepharidina*–*Searsia* and *Calotheca*–*Commiphora* pairs.

### 3.2. Areas of Endemism

The consensus map of the main areas of endemism identified by GIE and generally characterized by at least two synendemic co-generic (*Calotheca*) or co-subgeneric (*Afroblepharida* and *Blepharidina* s.str.) species is reported in Figure 9. For *Afroblepharida*, two main areas were identified in intertropical Africa: the first, with four synendemic species (*Blepharidina* (*Afroblepharida*) *afarensis* Biondi & D’Alessandro, *B*. (*A*.) *benadiriensis* Biondi & D’Alessandro, *B*. (*A*.) *somaliensis* (Bryant), *B*. (*A*.) *tajurensis* Biondi & D’Alessandro), corresponds to eastern Ethiopia and central Somalia and is largely attributable to the ecoregion “Somali *Acacia*-*Commiphora* bushlands and thickets ecoregion”; the second, with three synendemic species (*B*. (*A*.) *bantu* Biondi & D’Alessandro, *B*. (*A*.) *gedyei* (Bryant), *B*. (*A*.) *scripta* (Weise)), is located in central and southern Kenya and is mainly characterized by the ecoregion “Northern *Acacia*-*Commiphora* bushlands and thickets ecoregion”. In addition, three other small areas with only one endemic species were identified: north-central Nigeria (dominant ecoregion: “West Sudanian savanna”) with *B*. (*A*.) *zephyra* Biondi & D’Alessandro; the border area between Chad and Sudan (dominant ecoregion: “Sahelian *Acacia* savanna”) with *B*. (*A*.) *nubiana* Biondi & D’Alessandro; and central-eastern Sudan (dominant ecoregion: “Sahelian *Acacia* savanna”) with *B*. (*A*.) *antinorii* (Chapuis). For *Blepharidina*, two main areas were identified in sub-equatorial Africa: the first, with six synendemic species (*Blepharidina* (*Blepharidina*) *kasigauensis* D’Alessandro et al., *B*. (*B*.) *keniana* D’Alessandro et. al.; *B*. (*B*.) *knighti* (Bryant); *B*. (*B*.) *macarthuri* (Bryant), *B*. (*B*.) *ornaticollis* (Jacoby), *B*. (*B*.) *regalini* D’Alessandro et al.), is located between southern Kenya and north-eastern Tanzania (dominant ecoregion: “Northern *Acacia*-*Commiphora* bushlands and thickets”) and is widely overlapping with that of *Afroblepharida* in southern Kenya; and the second, with three synendemic species (*B*. (*B*.) *bimbiensis* (Bechyné), *B*. (*B*.) *carinata* (Bryant), *B*. (*B*.) *guttulata* (Baly)) corresponds to north-western Angola and south-eastern Democratic Republic of Congo (dominant ecoregion: “Western Congolian forest-savanna mosaic”). For *Calotheca*, two main areas were identified in southern Africa: the first, with four synendemic species (*Calotheca vittata* (Baly), *C*. *luteomaculata* D’Alessandro et al., *C*. *luteotessellata* D’Alessandro et al., *C*. *marmorata* (Baly)), is placed in the Drakensberg mountain area and is characterized by two dominant ecoregions, “Drakensberg montane grasslands, woodlands and forests” and “South Africa bushveld”; and the second, with seven synendemic species (*C*. *danielssoni* D’Alessandro et al.; *C*. *oberprieleri* D’Alessandro et al., *C*. *pallida* (Bryant), *C*. *parvula* (Weise), *C*. *prinslooi* D’Alessandro et al., *C*. *regularis* (Jacoby), *C*. *thunbergi* Biondi & D’Alessandro), is located in the Western Cape Province and is mainly characterized by “Succulent Karoo” and “Montane fynbos and renosterveld” ecoregions.

## 4. Discussion

The *Blepharida* genera complex (*Blepharida* s.l.) has a possible Gondwanan origin, with the separation of the close Malagasy genus *Xanthophysca* Fairmaire consequent to the break-up between Madagascar and eastern Africa (approximately 165–160 Ma), and of the genus *Calotheca* consequent to the separation between South America and Africa (approximately 130 Ma), confining the genus *Blepharida* Chevrolat s.str. in the American continent only [8,13,17,26,31,49]. No insights are currently available about the time origins of *Blepharidina* and *Afroblepharida*. Their separation from *Calotheca* in Africa was probably driven by changes in climate as adaptation to more xeric and warm environmental conditions, with a major occupancy of semidesert and savannah habitats [8]. This led to the current partially parapatric scenario, in which the suitable areas of *Calotheca* instead show a significant major occupancy of grasslands, shrublands, and forest habitats. Even the main areas of endemism identified for the three taxa show significant differences in their geographical location. Those of *Calotheca* are in two temperate zones (Drakensberg and Western Cape Province), while those of *Blepharidina* (southern Kenya/north-eastern Tanzania and north-western Angola/south-eastern Democratic Republic of Congo) and *Afroblepharida* (eastern Ethiopia/central Somalia and central/southern Kenya) are within the intertropical belt, with a wide overlap observed in southern Kenya.

The overall results of geostatistical analyses, whose applications are proven to give reliable insights about biodiversity-related issues [34,35,50], resulted in suggesting that both abiotic and biotic factors influenced the distribution of the target species and their co-occurrence. In fact, although abiotic factors seem to significantly affect and explain the distribution patterns of *Afroblepharida*, *Blepharidina* and *Calotheca*, the distribution of their host plants probably played an important role too [8]. Through its evolution, *Blepharida* s.l. has maintained a close relationship with its host plants, Anacardiaceae and Burseraceae [25,31]. These two plant families represent a pantropical sister pair, which have identical ages (origins in the Cretaceous age) and approximately the same number of species distributed on every continent except Antarctica. Unlike Anacardiaceae, which have shifted their climatic niches frequently during the past 100 million years, including the colonization of the temperate biomes, Burseraceae experienced fewer climatic niche expansions. In fact, there are no frost-tolerant species, and all Burseraceae are restricted to tropical and subtropical latitudes. Some of the lineage divergences in these two plant families may have been due to vicariance, but it is more probable that the great majority of them have been due to long-distance dispersal events and that both Anacardiaceae and Burseraceae have moved easily across the oceans [51].

The New World species of *Blepharida* mainly feed on Burseraceae and, to a lesser extent, on Anacardiaceae [13]. However, in Africa, *Calotheca* species are mainly associated with Anacardiaceae of the genus *Searsia*, and based on our analyses, the host plants of *Afroblepharida* and *Blepharidina* species are very probably Burseraceae of the genus *Commiphora*. This hypothesis is also well supported, beyond all our analyses, by the fact that the *Commiphora* species are common and even dominant in the ecoregions in which *Blepharidina* and, above all, *Afroblepharida*, are more present, such as “Northern *Acacia*-*Commiphora* bushlands and thickets” and “Somali *Acacia*-*Commiphora* bushlands and thickets”.

## 5. Conclusions

Our research reported that a multiple-scale and -predictor approach can give very useful clues for the distribution patterns of species which are possibly linked by an ecological relationship. The application of landscape-based methods, such as the Habitat specificity index, coupled with GIS techniques implemented through occurrence and species’ geographic information, resulted in a multifaceted pattern of results, showing, from different angles, the plant–host relationship we hypothesized. Additionally, the support offered by multivariate statistics and machine learning methods enabled the incorporation of additional predictors, thus encompassing both the biotic and abiotic components into our general framework of interpretation of the biogeography and ecology of the target taxa.

## Figures and Tables

**Figure 1 insects-12-00299-f001:**
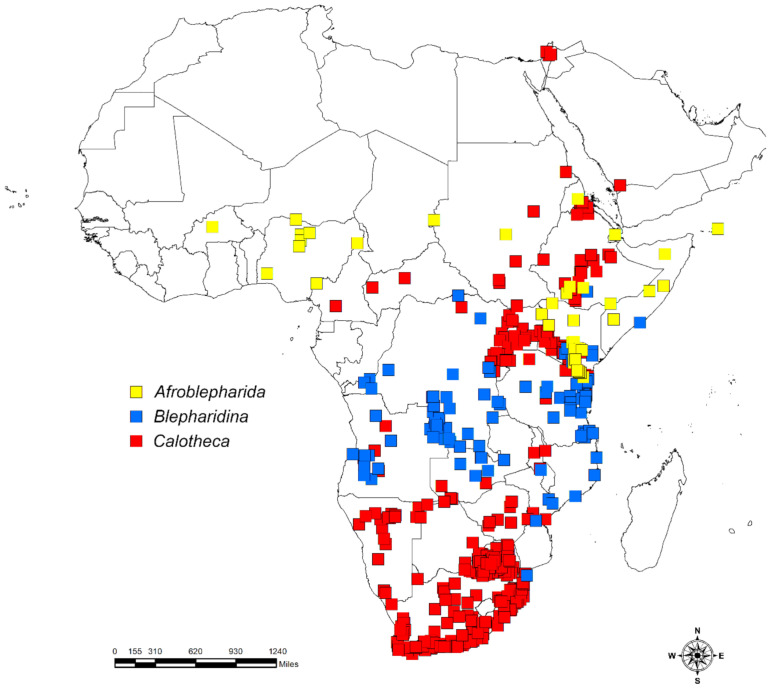
Distribution of the three target flea beetle taxa in the African continent and the Middle East.

**Figure 2 insects-12-00299-f002:**
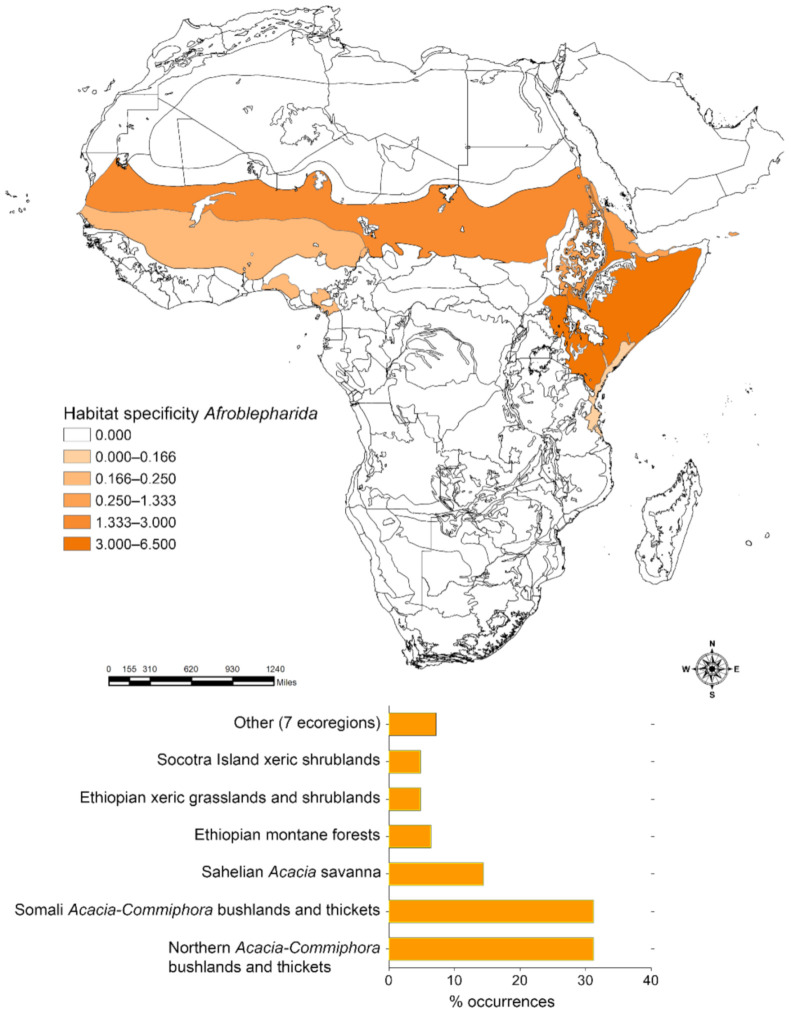
Habitat specificity applied to the Africa terrestrial ecoregions for *Afroblepharida* (**above**); number of occurrence localities (percent) falling within each ecoregion, with the length of the bar also proportional to the habitat specificity value (**below**).

**Figure 3 insects-12-00299-f003:**
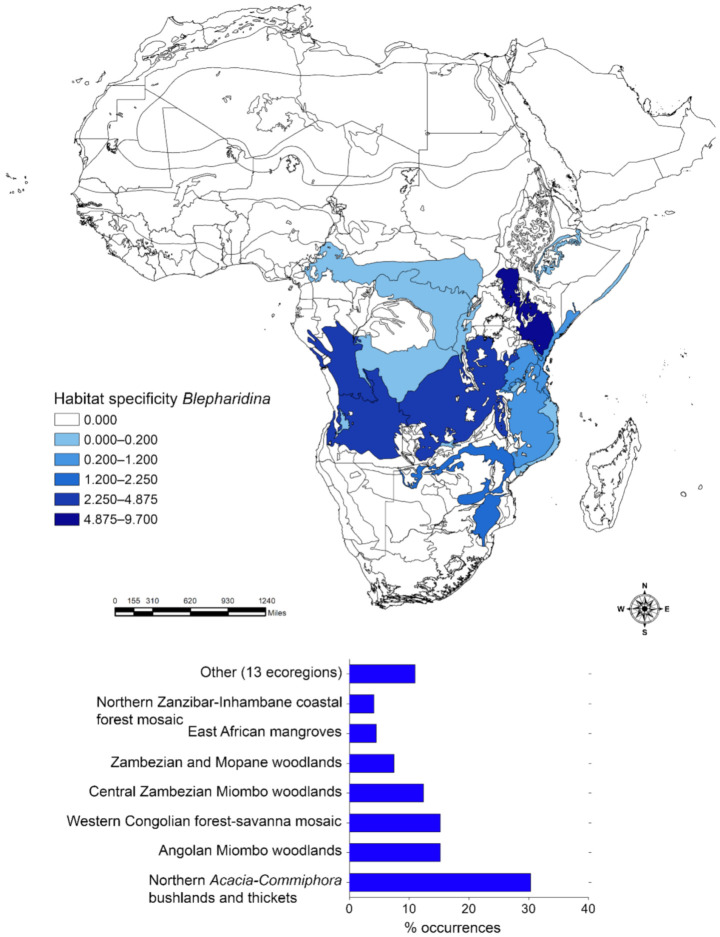
Habitat specificity applied to the Africa terrestrial ecoregions for *Blepharidina* (**above**); number of occurrence localities (percent) falling within each ecoregion, with the length of the bar also proportional to the habitat specificity value (**below**).

**Figure 4 insects-12-00299-f004:**
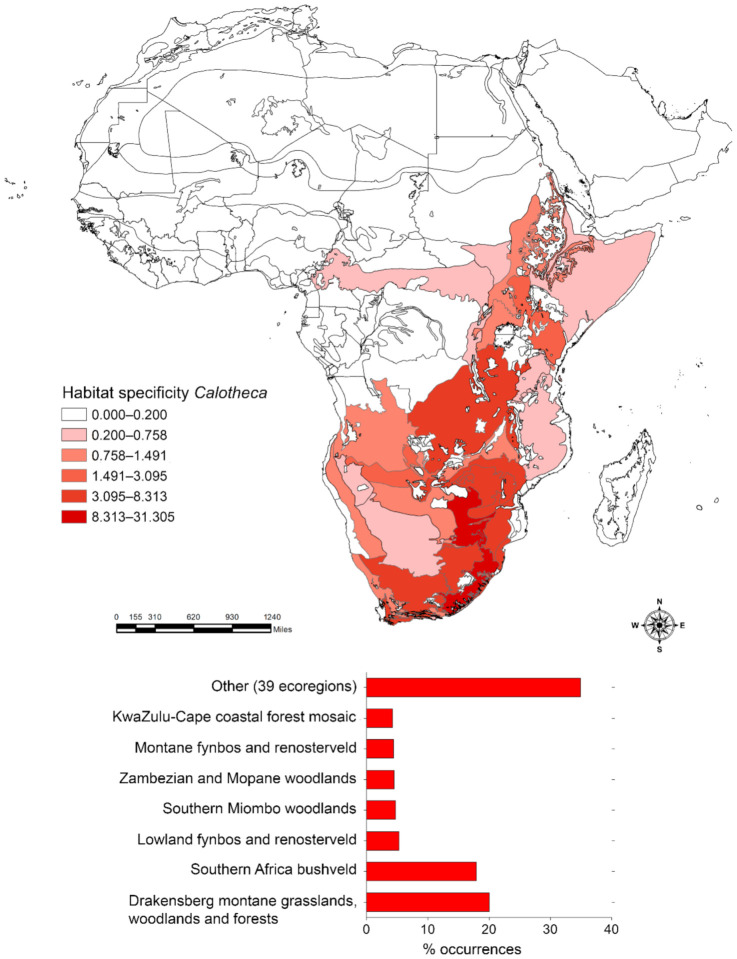
Habitat specificity applied to the Africa terrestrial ecoregions for *Calotheca* (**above**); number of occurrence localities (percent) falling within each ecoregion, with the length of the bar also proportional to the habitat specificity value (**below**).

**Figure 5 insects-12-00299-f005:**
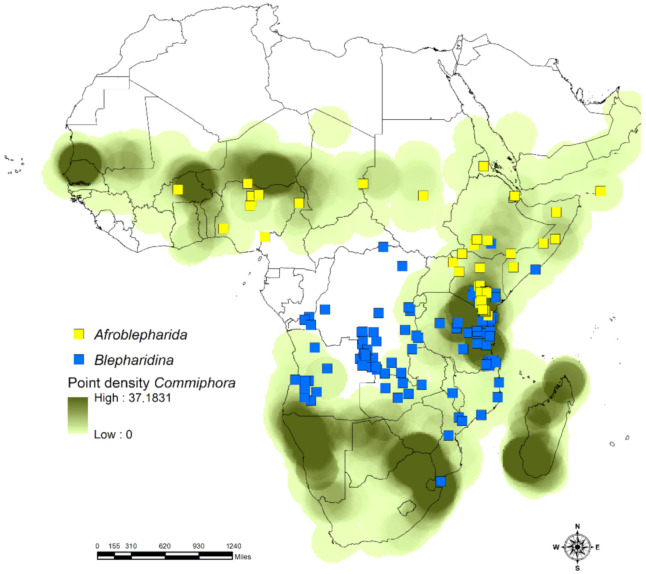
Occurrence localities of *Afroblepharida* and *Blepharidina* with point density of *Commiphora* plants; point densities for the two target flea beetle taxa are not displayed for graphical purposes.

**Figure 6 insects-12-00299-f006:**
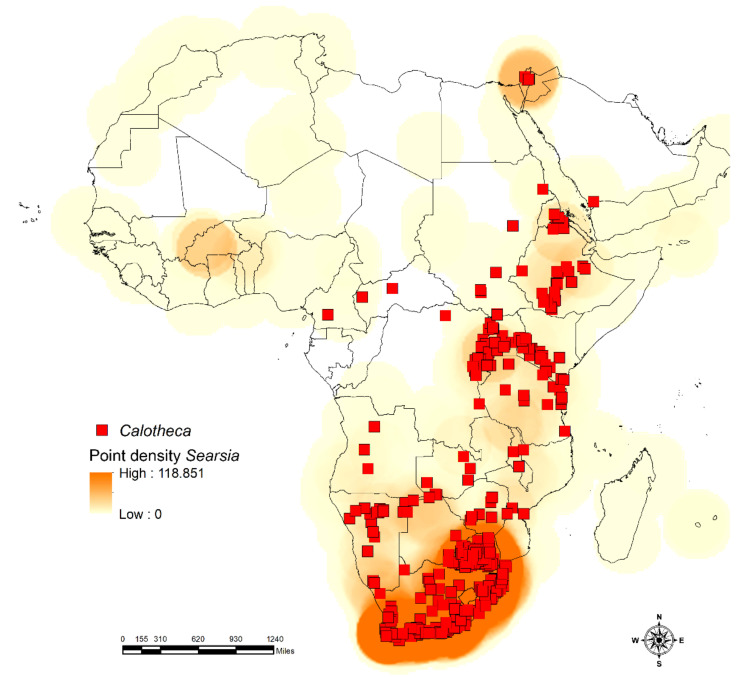
Occurrence localities of *Calotheca* with point density of *Searsia* plants; point densities for the target flea beetle taxon are not displayed for graphical purposes.

**Figure 7 insects-12-00299-f007:**
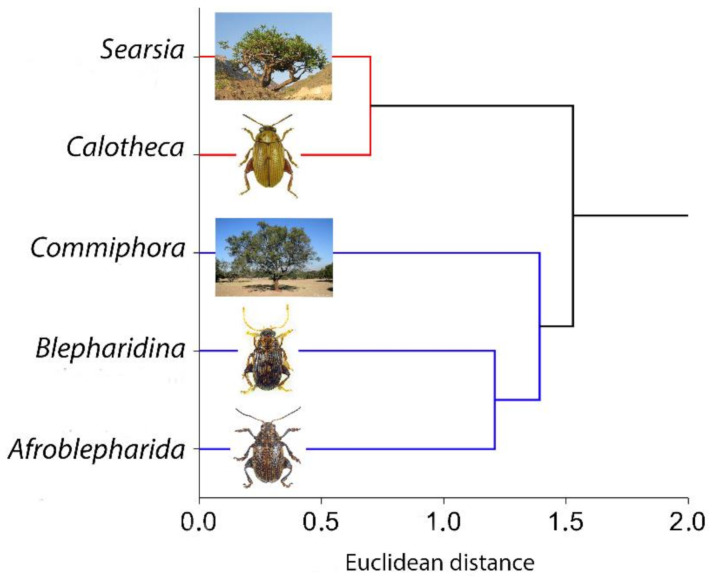
Dendrogram by cluster analysis (Euclidean distance/WPGMA) carried on the averages of the 19 Worldclim variables for *Afroblepharida*, *Blepharidina* and *Calotheca* flea beetles and *Searsia* and *Commiphora* plants (see text).

**Figure 8 insects-12-00299-f008:**
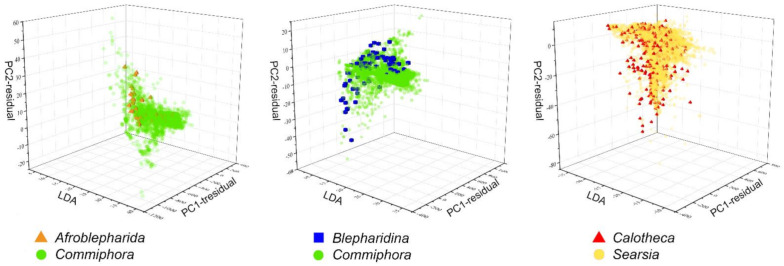
3D scatterplots for *Afroblepharida*–*Commiphora*, *Blepharidina*–*Commiphora* and *Calotheca*–*Searsia* pairs resulting from the niche overlap analysis, based on climatic predictors, inferred through the Hyperoverlap R package [46].

**Figure 9 insects-12-00299-f009:**
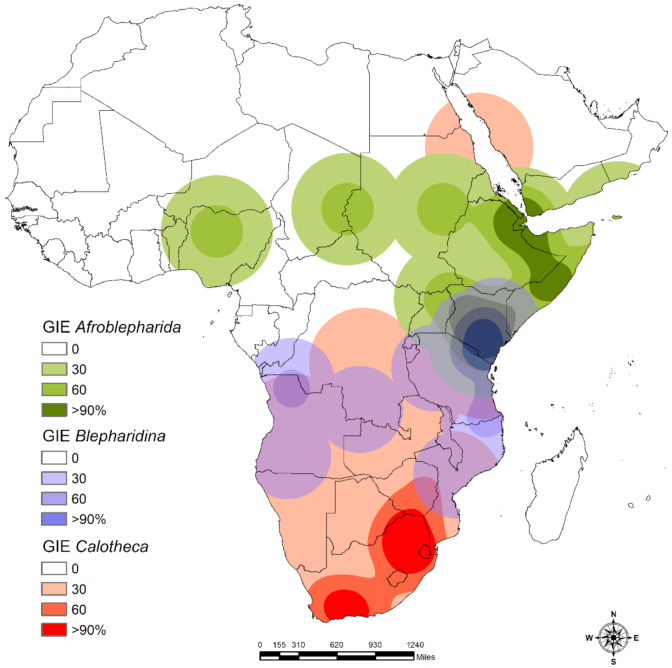
Areas of endemism of *Afroblepharida*, *Blepharidina* and *Calotheca* resulting from the Geographic Interpolation of Endemism (GIE) analysis inferred in the GIS environment.

**Table 1 insects-12-00299-t001:** Averages of the 19 Wordclim variables calculated for all occurrences of every flea beetle and plant taxon. BIO1: Annual Mean Temperature; BIO2: Mean Diurnal Range; BIO3: Isothermality [(BIO2/BIO7)*100]; BIO4: Temperature Seasonality (Standard Deviation); BIO5: Max. Temperature of Warmest Month; BIO6: Min. Temperature of Coldest Month; BIO7: Temperature Annual Range (BIO5-BIO6); BIO8: Mean Temperature of Wettest Quarter; BIO9: Mean Temperature of Driest Quarter; BIO10: Mean Temperature of Warmest Quarter; BIO11: Mean Temperature of Coldest Quarter; BIO12: Annual Precipitation; BIO13: Precipitation of Wettest Month; BIO14: Precipitation of Driest Month; BIO15: Precipitation Seasonality (Coefficient of Variation); BIO16: Precipitation of Wettest Quarter; BIO17: Precipitation of Driest Quarter; BIO18: Precipitation of Warmest Quarter; BIO19 Precipitation of Coldest Quarter.

Variables	*Afroblepharida*	*Blepharidina*	*Calotheca*	*Searsia*	*Commiphora*
BIO1 (°C)	24.40	23.07	19.35	17.99	23.81
BIO2 (°C)	11.41	11.27	12.86	12.53	13.13
BIO3 (%)	68.83	66.89	61.32	57.43	61.61
BIO4 (°C)	157.48	151.94	299.71	337.97	281.59
BIO5 (°C)	33.04	31.12	29.00	28.17	33.73
BIO6 (°C)	16.22	14.07	7.53	6.12	12.06
BIO7 (°C)	16.82	17.05	21.47	22.05	21.68
BIO8 (°C)	24.87	23.83	21.28	19.29	25.59
BIO9 (°C)	23.05	21.43	16.30	15.70	20.74
BIO10 (°C)	26.30	24.61	22.58	21.78	26.83
BIO11 (°C)	22.43	20.98	15.38	13.60	19.99
BIO12 (mm)	667.36	1130.15	747.79	693.88	560.66
BIO13 (mm)	159.74	228.47	132.35	120.40	133.12
BIO14 (mm)	7.19	8.28	12.83	13.35	4.14
BIO15 (%)	96.83	87.83	71.22	66.86	106.76
BIO16 (mm)	344.36	575.02	349.32	326.07	332.65
BIO17 (mm)	29.36	36.01	48.71	47.84	16.50
BIO18 (mm)	184.91	305.68	265.82	239.17	211.67
BIO19 (mm)	60.39	61.76	88.54	103.94	34.83

## Data Availability

The datasets for the flea beetles *Afroblepharida*, *Blepharidina*, and *Calotheca*, generated using both known localities and original data, can be requested from the authors. The datasets of the plants *Searsia* and *Commiphora*, obtained from the GBIF portal (*Commiphora* accessed on 27 January 2021 [38]; *Searsia* accessed on 13 January 2021 [39]) and the available literature and checklists of African countries, can also be requested from the authors. The ecoregion shapefile was downloaded from http://www.fao.org/land-water/land/land-governance/land-resources-planning-toolbox/category/details/en/c/1036295/ (accessed on 27 January 2021).

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
