# Peer review of "Habitat Specificity, Host Plants and Areas of Endemism for the Genera-Group Blepharida s.l. in the Afrotropical Region (Coleoptera, Chrysomelidae, Galerucinae, Alticini)"

_insects, 2021, doi:10.3390/insects12040299_

Round 1
Reviewer 1 Report
This is an interesting predictive, model-based analysis of a group of Afrotropical flea beetles. It is quite theoretical, but a good use of some modelling systems that have predictive value for habitat specificity using mainly biogeographic/distributional information of the taxa involved.
The use of the words "host plants" in the title may a little misleading, because, as far as I can tell there are relatively few species in these genera for which the host plants are definitively known and this fact is not really mentioned anywhere in this manuscript.
The first sentence of the Discussion section that says that Blepharida (sensu lato) has a Gondwanan origin. This seems particularly and possibly overly speculative, especiallybecause, as stated here by the authors "No insights are currently available about the time of origin of Blepharidina and Afroblepharidina"!! And that Calotheca and Blepharida were separated by the separation of South America (with no Blepharida species) and Africa.
Again, in the last paragraph of the Discussion, they say that "..host plants of Afroblepharidina and Blepharidina are"..very probably Burseraceae of the genus Commiphora." indicates the lack of validated host records, thus it is very speculative. And in the Conclusion they say "..plant host relationship we hypothesize..". So, I am a little uncomfortable with the impression in the Title, and elsewhere, that host plants are a significant part of this analysis. It is true that for a relatively small number of the species in these three genera, there are some some host records, so I assume this speculation is based on those, but this needs to be better explained near the beginning, possibly in the Introduction.
There are multiple places where scientific names are not italicized and should be. I have indicated these on the attached reviewed manuscript, including in Figures 2 and 6, Acacia, Calotheca, etc.

Reviewer 2 Report
This paper gives important ecological information about some species representatives of an important beetle family, using appropriate informatic tools.
In the overall the paper is well designed, performed and presented and I could suggested just very minor notes reported in the text.
I have just a couple of curiosities:
I would guess that a crucial aspect of this analysis is an accurate and extensive (both spatially and temporally) sampling of the beetles. Could it be useful to give a quick summary about the quality of the samplings (i.e. if there are areas never or poorly studied, seasonal gaps, etc)?
On the other hand I'm a bit surprised that for the species of Blepharidina s.l, despite the numerous samples, there is a lack of ecological information, above all about host species. Were the specimens mainly collected by traps or sweeping? Could be useful (in future or for other problematic associations) to do molecular analysis of the gut content?
